# In-Cell Labeling Coupled to Direct Analysis of Extracellular Vesicles in the Conditioned Medium to Study Extracellular Vesicles Secretion with Minimum Sample Processing and Particle Loss

**DOI:** 10.3390/cells11030351

**Published:** 2022-01-20

**Authors:** Anissa Viveiros, Vaibhavi Kadam, John Monyror, Luis Carlos Morales, Desmond Pink, Aja M. Rieger, Simonetta Sipione, Elena Posse de Chaves

**Affiliations:** 1Department of Pharmacology, Faculty of Medicine and Dentistry, University of Alberta, Edmonton, AB T6G 2H7, Canada; aviveiro@ualberta.ca (A.V.); vaibhavi@ualberta.ca (V.K.); monyror@ualberta.ca (J.M.); lmorales@ualberta.ca (L.C.M.); 2Neuroscience and Mental Health Institute, Faculty of Medicine and Dentistry, University of Alberta, Edmonton, AB T6G 2H, Canada; 3Department of Oncology, Faculty of Medicine and Dentistry, University of Alberta, Edmonton, AB T6G 2H7, Canada; desmond.pink@nanosticsdx.com; 4Department of Microbiology and Immunology, Faculty of Medicine and Dentistry, University of Alberta, Edmonton, AB T6G 2H7, Canada; aja@ualberta.ca

**Keywords:** extracellular vesicles, fluorescent labeling, imaging flow cytometry, ultracentrifugation, size exclusion chromatography, nanoparticle tracking analysis

## Abstract

Extracellular vesicles (EVs) are involved in a multitude of physiological functions and play important roles in health and disease. The largest proportion of studies on EVs is based on the analysis and characterization of EVs secreted in the cell culture medium. These studies remain challenging due to the small size of the EV particles, a lack of universal EV markers, and sample loss or technical artifacts that are often associated with EV labeling for single particle tracking and/or separation techniques. To address these problems, we characterized and validated a method for in-cell EV labeling with fluorescent lipids coupled with direct analysis of lipid-labeled EVs in the conditioned medium by imaging flow cytometry (IFC). This approach significantly reduces sample processing and loss compared to established methods for EV separation and labeling in vitro, resulting in improved detection of quantitative changes in EV secretion and subpopulations compared to protocols that rely on EV separation by size-exclusion chromatography and ultracentrifugation. Our optimized protocol for in-cell EV labeling and analysis of the conditioned medium reduces EV sample processing and loss, and is well-suited for cell biology studies that focus on modulation of EV secretion by cells in culture.

## 1. Introduction

Extracellular vesicles (EVs) are cell-derived particles delimited by a membrane. They are shed by most cells and play pivotal roles in intercellular communication and signaling both in health and disease conditions [1,2,3,4]. Further to this, EVs can carry cancer biomarkers and therefore have diagnostic and prognostic value [5,6]. There is also a growing appreciation of the utility of EVs as therapeutics [7,8,9,10], gene editing tools [11], drug carriers [12], and in other clinical applications [13,14]. Owing to this multitude of physiological functions and therapeutic applications, the interest in EVs has grown exponentially in the past few years and recent technological advances have considerably expanded the tools available for EV studies. Nevertheless, the small and heterogeneous size (30–1000 nm) of EVs and the absence of universal EV markers are still challenges that significantly hamper studies on the biology of EVs, with many aspects related to biogenesis, secretion, and biology still remaining unclear [15,16].

Recent surveys indicated that the largest proportion of studies on EVs are based on the analysis of EVs secreted in the cell culture conditioned medium [17,18]. The most common method of EV separation is ultracentrifugation alone or combined with other techniques. Ultrafiltration-size exclusion chromatography (SEC) has been gaining popularity in the past 5 years [18]. A variety of methods are used to characterize EVs including Western blotting, electron microscopy, single particle tracking (mostly nanoparticle tracking analysis (NTA)), and flow cytometry [17,18].

To gain deeper insights into the mechanisms that underlie EV secretion and other aspects of EV biology, improved methods for EV separation and analysis are required. Characterization of EV subpopulations that might express different markers or carry different cargos is also of paramount importance, as biochemical analysis of bulk EV preparations may not adequately reveal quantitative and qualitative differences in EV subpopulations. Significant strides in this direction have been made through the development of methods for single particle analysis by high-resolution flow cytometry [19] and imaging flow cytometry (IFC). A comprehensive analysis for applying this technology to the study of EVs is available [20,21,22]. Characterization of EVs by IFC requires the use of fluorescent markers because EVs display a very low level of scatter and most cannot be detected as brightfield (BF) images because the resolution of the BF camera is not sufficient [22]. EV labeling for these studies is most often performed after separation from conditioned medium or biological fluids, an approach that presents us with several challenges, including elimination of fluorescently-labeled non-EV particles and/or free-dye [23,24,25]. Moreover, although several dyes are already in use for EV labeling, there is a recognized need for improved dyes and labeling methods [26].

Here, we aimed to design a simplified and reliable method to study EV secretion and regulation in cell models. In this study, (i) we present a method to label EVs that consists of labeling parental cells with lipophilic cationic indocarbocyanine dyes prior to EV collection and analysis by IFC; (ii) we demonstrate that the cleared conditioned medium (depleted of cell debris and apoptotic bodies) represents the preferable sample for analysis of EV secretion, with minimum processing and loss compared to EV preparations that rely on ultracentrifugation or size exclusion chromatography; and (iii) we provide evidence that, at least in some cases, EV analysis performed in the cleared conditioned medium allows detection of changes in EV secretion that cannot be distinguished if the samples undergo ultracentrifugation.

## 2. Materials and Methods

### 2.1. Materials

Opti-MEM^®^ reduced-serum medium, Dulbecco’s Modified Eagle Medium (DMEM), penicillin/streptomycin, fetal bovine serum (FBS), N-2 supplement, B27 supplement, and Geneticin^®^ were from Gibco (Thermo Fisher Scientific, Waltham, MA, USA). Protease inhibitor cocktail tablets were from Roche (Basel, Switzerland). Clarity™ Western ECL Substrate and nitrocellulose membranes were purchased from Bio-Rad (Hercules, CA, USA). Bicinchoninic acid (BCA) protein assay kit was from Thermo Fisher Scientific (Waltham, MA, USA). Vybrant™ DiI was from Invitrogen (Thermo Fisher Scientific, Waltham, MA, USA). All horseradish peroxidase-conjugated secondary antibodies were from Amersham Biosciences (Cytiva Life Sciences, Marlborough, MA, USA). The qEV columns (SP1) were from Izon Science (Christchurch, New Zealand). Immobilon PVDF membranes, Amicon Ultra-15 100K MWCO, Amicon Ultra-4 10K MWCO and Amicon Ultra-0.5 10K MWCO filters, NP-40 (IGEPAL^®^ CA-630) and Bafilomycin A1 were purchased from Millipore Sigma (Burlington, MA, USA). Glutathione Sepharose 4B beads were purchased from GE Healthcare (Chicago, IL, USA). Penicillin/streptomycin were purchased from HyClone™ (GE Healthcare, Chicago, IL, USA). Pre-lubricated pipette tips (Maxymum Recovery™, Axygen^®^, 1–200 μL and 100–1000 μL) and microcentrifuge tubes were purchased from Corning Life Sciences (Corning, NY, USA).

### 2.2. Cell Culture

Mouse neuroblastoma Neuro2a (N2a) cells, obtained from ATCC (CCL-131™), were cultured in DMEM/Opti-MEM I (1:1) supplemented with 5% heat-inactivated fetal bovine serum (FBS) and 1% penicillin/streptomycin. Cells were maintained at 37 °C with 5% CO_2_ and split every 3–4 days. Medium was changed every second day. HeLa cells, obtained from ATCC (CRM-CCL-2™), were cultured in DMEM supplemented with 10% heat-inactivated fetal bovine serum (FBS), 2 mM L-glutamine, and 0.11 g/L sodium pyruvate. Cells were maintained at 37 °C with 5% CO_2_ and split every 2–3 days.

### 2.3. Fluorescent Labeling of Cells for EV Detection

Cells were labeled with the lipophilic membrane stain DiI (1,1′-dioctadecyl-3,3,3’,3’-tetramethylindocarbocyanine perchlorate; λEx/λEm = 549/565 nm) or its longer wavelength analogue DiD (1,1′-dioctadecyl-3,3,3′,3′-tetramethylindodicarbocyanine, 4-chlorobenzenesulfonate salt; λEx/λEm = 644/665 nm) according to the manufacturer’s instructions (Invitrogen, Thermo Fisher Scientific, Waltham, MA, USA). Briefly, cells were detached with 0.25% trypsin-EDTA, resuspended in serum-free medium (Opti-MEM:DMEM (1:1), 1% penicillin/streptomycin, 2 mM L-glutamine and 0.11 g/L sodium pyruvate) to a concentration of 1 × 10^6^ cells/mL. Five μL of the dye solution (1 mM) was added to each 1 mL of cell suspension and incubated for 20 min at 37 °C in the dark and then centrifuged at 300× *g* for 5 min at r.t. The stained cell pellet obtained was further subjected to three rounds of centrifugation in growth media to remove unbound dye. Cells were plated at 1.8 × 10^6^ cells/10 cm diameter plate in DMEM/Opti-MEM I (1:1) supplemented with 5% FBS, 1% penicillin/streptomycin, 2 mM L-glutamine, and 0.11 g/L sodium pyruvate, and maintained in culture for 18 h before starting EV collection. When using dishes of other sizes, the seeding densities were determined such that the confluence of the cells at the time of harvesting was around 80%. For EV collection, the medium was replaced with a medium containing N-2 supplement, but no FBS (serum- and phenol red-free medium), and the conditioned medium was harvested after 24 h.

### 2.4. Collection of EVs in Cell-Conditioned Media

After cell treatment, medium was replaced with serum-free media supplemented with N-2 supplement (OptiMEM:DMEM 1:1, 1% penicillin/streptomycin, 2 mM L-glutamine and 0.11 g/L sodium pyruvate with 1× N-2 for N2a cells, and DMEM 1% penicillin/streptomycin, 2 mM L-glutamine and 0.11 g/L sodium pyruvate with 1× N-2 for HeLa cells), and allowed to be conditioned by cells for 24 h. The conditioned medium was collected in polypropylene tubes and centrifuged at 2000× *g* for 10 min at 4 °C in an Eppendorf^®^ Centrifuge 5810 R (Eppendorf AG, Hamberg, Germany), using an A-4-81 swinging bucket rotor, to pellet any remaining cells, apoptotic bodies, and cell debris, resulting in a cleared conditioned medium. The cells were harvested in 500 μL of PBS and lysed by sonication. Protein concentration in cell lysates was measured with a Pierce™ BCA Protein Assay and or a Pierce™ Enhanced BCA Protein Assay (Thermo Fisher Scientific, Waltham, MA, USA) according to the manufacturer’s instructions. Before further processing, cleared conditioned media from different samples were adjusted based on cellular protein content to ensure that all cleared conditioned media were derived from the same amounts of cellular proteins. EV samples were always kept on ice. They were mixed by tube inversion or by gently pipetting up and down and analyzed immediately. All culture media and diluents were freshly prepared and filtered through a 0.1 μm filter.

### 2.5. Fluorometry

DiI fluorescence was measured in cell lysates and in the cleared conditioned medium using a SpectraMax^®^ i3x multi-mode microplate reader (Molecular Devices, San Jose, CA, USA). λEx/λEm is 540/580 nm for DiI and 644/674 nm for DiD. The bandwidth of all excitation and emission wavelengths was set to 15 nm. Fluorescence was measured in 96-well black-bottom plates by well-scan reading. Fifty microliters of the sample were loaded in each well in triplicate.

### 2.6. Separation of EVs by Sequential Ultracentrifugation

EV separation by sequential ultracentrifugation (UC) was performed in accordance with previously described protocols [27], with some modifications. Briefly, the cleared conditioned medium was centrifuged in an Optima™ MAX-XP Ultracentrifuge (Beckman Coulter, Brea, CA, USA) at 100,000× *g* at 4 °C for 90 min, using a MLA-55 fixed-angle rotor (k-factor: 53). The supernatant (Sup_100K_) was set aside at 4 °C and the pellet was washed with 1 mL of phosphate buffered saline (PBS) and centrifuged again at 100,000× *g* at 4 °C for 90 min, in a MLA-130 fixed-angle rotor (k-factor: 8.7). The resulting pellet (Pellet_100K_) was resuspended in 50–100 μL PBS. EVs were kept at 4 °C and analyzed immediately following resuspension.

### 2.7. Separation of EVs by Ultrafiltration Combined with Size Exclusion Chromatography

Cleared conditioned media were concentrated using Amicon^®^ Ultra-15 Centrifugal Filters (10,000 MWCO or 100,000 MWCO) (Millipore Sigma, Burlington, MA, USA). The clear conditioned medium was loaded onto the filter and concentrated by centrifugation at 2608× *g* at 4 °C to ≤ 300 μL. The concentrate was collected, and the filter membrane was washed with 100 µL serum-free medium, which was added to the concentrate. Where necessary, sample volume was adjusted to 550 μL with serum-free medium. A 50 μL aliquot was used for DiI or DiD measurements by spectrofluorometry and the remaining 500 μL were used for size exclusion chromatography (SEC). For the separation of EVs, qEV original size (10 mL) exclusion columns (Izon Science, Christchurch, New Zealand) were used as indicated by the manufacturer. SEC columns were stored at 4 °C in PBS containing 0.05% sodium azide and used according to the manufacturer’s instructions. After column equilibration at r.t., the elution time of 10 mL of PBS was recorded to ensure optimal column packing and performance. Twenty-two fractions of 0.5 mL were collected using PBS as eluent. The presence of EVs in the various fractions was determined by measuring DiI or DiD fluorescence (λEx/λEm = 540/580 nm and 644/674 nm respectively) in each fraction. Protein content of each fraction was determined by measuring the absorbance at 280 nm using Nanodrop™ 2000c (Thermo Fisher Scientific, Waltham, MA, USA). The EV-rich, protein-low fractions (generally between fractions 5 and 11 included) referred to as “EV fractions” were pooled and concentrated using Amicon^®^ Ultra-2mL filters (10,000 MWCO) (Millipore Sigma (Burlington, MA, USA) that were blocked with Tween-80 (Millipore Sigma (Burlington, MA, USA) as previously described [28]. EVs were maintained at 4 °C and analyzed immediately.

### 2.8. In Vitro Labeling of EVs

Labeling of EVs in vitro was performed by adding fluorescently-conjugated antibodies and Annexin V directly to the cleared conditioned medium or to the Pellet_100K_ resuspended in PBS. For CD9 detection we used Alexa Fluor^®^ 647-conjugated anti-CD9 antibody (BioLegend, Cat#124810, San Diego, CA, USA) at a final concentration of 2.5 μg/mL, and we ran a separate sample with IgG isotype control (Cat#400526) (BioLegend, San Diego, CA, USA) in parallel to ensure the absence of non-specific binding. All antibodies were centrifuged for 10 min at 14,000× *g* before use. Pacific Blue™ Annexin V (Invitrogen, Cat#A35122) (Thermo Fisher Scientific, Waltham, MA) was used to detect surface phosphatidylserine. Briefly, samples from the same preparation of cleared conditioned medium and Pellet_100K_ were diluted 1:1 with 2X Annexin V binding buffer (50 μL of each for a total volume of 100 μL). Next, we added 5 μL of Annexin V (1:20 dilution, concentration proprietary) and 5 μL of a 1:10 dilution of anti-CD9 in PBS. Samples were incubated at r.t. for 15 min, in the dark, prior to direct analysis by IFC. Controls including only the Annexin V label, or the CD9 antibody, or DiI label were used to create a compensation matrix as described in the next section. Titrations of CD9 antibody and Annexin V were performed to determine the appropriate concentrations (Appendix A).

### 2.9. Analysis of EVs by Image Flow Cytometry (IFC)

All samples were analyzed on an Amnis ImageStreamX MkII instrument (Luminex Corporation, Austin, TX, USA) equipped with four lasers (120.00 mW 405 nm, 200.00 mW 488 nm, 150.00 mW 642 nm, and 70.00 mW 785 nm). All lasers for fluorophore-labeled EV detection were set to maximum powers. All data were acquired using a 60× magnification objective, with numerical aperture of 0.9. The 60× magnification generates the lowest pixel resolution (0.3 µm^2^/pixel) and will also set the core stream width to 7 μm [21,22]. All samples were collected in the following channels: Ch03 (560–595 nm) for Di, Ch07 (435–505 nm) for Annexin V, Ch11 (642–745 nm) for DiD or CD9. Ch01 (435–480 nm) was used for brightfield imaging and Ch06 (745–780 nm filter) for SSC detection. Where indicated, the 405 laser (Ch07) was also used for SSC detection. Standard, unfiltered BioSure sheath fluid (D-PBS, pH 7.4) was used for all measurements. For each sample, acquisition was set up to capture all events that displayed lower SSC than the SpeedBeads (Amnis SpeedBead^®^ Kit) (Luminex Corporation, Austin, TX, USA). Unstained EVs secreted by unlabeled cells were used to assure that no fluorescence signal was detected in any channel. Similarly, the buffer only and the unconditioned N-2-supplemented medium were run through IFC to determine the background signal in each experiment. In all cases, minimal to no signal events were detected. In experiments in which EVs subtypes were analyzed, single stained controls were always run in parallel to be able to establish a compensation matrix. In addition, controls with single and double antibodies or antibodies/AnnexinV in buffer were included. All samples were subjected to detergent lysis with NP-40 (0.5%) for 30 min at r.t., as described previously [21,29].

Data analysis was performed using Amnis IDEAS software (version 6.2) (Luminex Corporation, Austin, TX, USA). We gated on Intensity_Ch06 (SSC) in order to remove any remaining SpeedBeads from the analysis. To ensure we were analyzing only 1 EV in each image, we created a mask to identify DiI intensity (Intensity (M03,Ch03_DiI, 50-4095)) or DiD intensity (Intensity(M11,Ch11_DiD, 60-4095)). Using this mask, we developed a spot count feature (Spot Count_Intensity (M03,Ch03_DiI, 50-4095)_4) or (Spot Count_Intensity (M11,Ch11_DiD, 60-4095)_4) and gated on images that had no more than 1 DiI or DiD spot. The resulting population represented EVs.

In experiments in which DiI-stained EVs were labeled in vitro with antiCD9 antibodies and/or AnnexinV, a compensation matrix for spectral spillover was calculated using the single stained controls and the IDEAS_6.2 compensation wizard. The matrix was applied to all other samples, including the buffer controls, to remove spectral crosstalk between fluorochrome channels. These compensated files are then further analyzed using a variety of data analysis tools available in the IDEAS and FCS Express software v7 (DeNovo Software). The log of the intensity feature within the combined mask (MC) of all channels is used to plot both fluorescence and scatter parameters. The intensity feature is the sum of all pixel values within the mask minus the background. IDEAS allows all feature values to be exported as .fcs files, which can be analyzed by other flow cytometry software programs.

### 2.10. Nanoparticle Tracking Analysis (NTA)

Measurement of particle size and concentration was performed by NTA, using a NanoSight LM10 system equipped with a 405 nm laser (NanoSight, Amesbury, UK). EV samples were diluted with 0.1 μm-filtered PBS to achieve a concentration of 20–100 particles/frame and injected into the sample chamber with sterile syringes. Five aliquots from each sample were measured, each run for 1 min. The precise temperature during sample acquisition was recorded manually to accurately determine particle concentration. All samples were captured with the same camera level and detection threshold (camera level set at 14 and detection level threshold set at 7). Instrument settings were checked prior to data collection using NIST traceable 200 nm-polystyrene beads (3000 series) (Thermo Scientific, Waltham, MA, USA) diluted in 10mM potassium chloride. Comparison of particle concentration across different samples used dilution- and volume-corrected values.

### 2.11. Transmission Electron Microscopy

Fifty microliter aliquots of the EV pellet (Pellet_100K_, freshly separated by UC and resuspended in PBS) and SEC fractions were fixed with an equal volume of 2× Karnovsky fixative (0.2M Na cacodylate, 4% paraformaldehyde and 4% glutaraldehyde) (Ted Pella Inc., Redding, CA, USA), mixed gently and incubated on ice for 30 min. Five to ten µL of this mix were transferred onto carbon-formvar-coated grids (Ted Pella Inc., Redding, CA, USA) following glow discharge. After 5–10 min, the grids were washed gently and transferred on top of 2.5% uranyl acetate (Electron Microscopy Sciences, Hatfield, PA, USA) for 10 min. The grids were washed gently again and left to dry on a Kimwipe^®^ in the dark. Imaging was performed using a JEM-2100 microscope (JOEL Ltd., Akishima, Tokyo, Japan).

### 2.12. Immunoblotting

Cells were harvested in RIPA buffer (140 mM NaCl, 20 mM Tris, 1% SDS, 0.1% NP40, 0.5% sodium deoxycholate, pH 7.4) supplemented with protease inhibitor cocktail and sonicated. Protein content was determined using the Pierce™ Enhanced BCA Protein Assay (Thermo Fisher Scientific, Waltham, MA, USA) according to manufacturer’s instructions. Absorbance at 562 nm was measured using a SpectraMax^®^ i3x Multi-Mode Microplate Reader (Molecular Devices, San Jose, CA, USA). Proteins were resolved by SDS-PAGE in 16% polyacrylamide gels and transferred to PVDF membranes overnight at 4 °C. Membranes were washed three times with TBS + 0.1% Tween 20 (TTBS) and blocked for 1 h in TTBS containing 5% non-fat milk. Membranes were probed overnight at 4 °C with the following primary antibodies: anti-ALIX (AIP1) (1:250; Cat#611621) (BD Biosciences, Franklin Lakes, NJ, USA), anti-CD9 (1:1000; Cat#ab92726) (Abcam, Cambridge, England, UK), anti-LC3 (1:500; Cat#NB100-2220) (Novus Biologicals, Littleton, CO, USA), and anti-actin (1:2000; Cat#4957S) (Cell Signaling Technology, Danvers, MA, USA). All primary antibodies were diluted in TTBS containing 5% bovine serum albumin (BSA), with the exception of the anti-LC3 and anti-actin antibodies, which were diluted in blocking buffer. The next day, membranes were washed twice with TBS, twice with TTBS and twice again with TBS followed by 1 h incubation with the appropriate secondary antibodies (1:2000 in blocking buffer) at r.t. with gentle agitation. Membranes were washed again twice for 5 min each with TBS, TTBS, and TBS respectively. Immunoreactivity was detected with Clarity™ Western ECL Substrate (Bio-Rad, Hercules, CA, USA) and visualized with a Li-COR C-DiGit western blot scanner (LI-COR Biosciences, Lincoln, NE, USA). Band densitometry was performed using Image Studio™ Lite (version 5.2.5) (LI-COR Biosciences, Lincoln, NE, USA).

### 2.13. Statistical Analysis

Statistical analyses were performed in GraphPad PRISM (version 9.1.2) (GraphPad Software Inc., San Diego, CA, USA). Comparisons between two samples were performed using paired or unpaired Student’s *t*-test. For multiple comparisons, one-way ANOVA was applied. Multiple comparisons posthoc testing was performed as indicated in the figure legends.

## 3. Results

### 3.1. Cell and EV Labeling

To minimize variability due to sample handling, we developed a method to label EVs prior to their secretion by incubating parental cells with the lipophilic cationic indocarbocyanine dye DiI (Figure 1). Cell analysis by imaging flow cytometry (IFC) demonstrates that the dye is incorporated throughout the cells and labels the plasma membrane as well as intracellular membranes, up to at least 46 h from the initial labeling (Figure 2A). The fluorescence of stained cells was around two orders of magnitude above background autofluorescence (Figure 2B,C). Over 95% of cells were labeled with DiI up to 46 h post staining and throughout the timeframe of EVs collection into the conditioned medium (Figure 2C). The overall median cell fluorescence had decreased at the latest timepoint measured (46 h) compared to cells freshly stained or 22 h post staining (Figure 2D). However, we could not detect any change of the median EV fluorescence in the timeframe of EV collection (Figure 2E). Similar results were obtained when N2a cells were labeled with DiD instead of DiI (Appendix A). The labeling method was applied successfully to other cells in culture, including PC12 cells, striatal cells from mutant huntingtin knock-in mice, primary human fibroblasts, immortalized astrocytes expressing APOE3, and HeLa cells (Appendix A).

The conditioned medium containing EVs secreted by DiI-stained N2a cells was cleared from cell debris and apoptotic bodies by centrifugation at 2000× *g* for 10 min (cleared conditioned medium) and directly analyzed by IFC, or further separated by ultracentrifugation or ultrafiltration-size exclusion chromatography (Figure 1). IFC has recently emerged as a powerful technique for single particle analysis of fluorescently-labeled EVs [21,22]. EVs secreted by DiI-stained cells were amenable to single particle analysis by IFC. The majority of the low scatter events detected in the cleared conditioned medium (82.7 ± 15.0%) were DiI-positive (Figure 3A). Moreover, 98.8 ± 0.1% of the DiI-positive events were detected as single objects by IFC. DiI-positive events were highly sensitive to lysis by the detergent NP-40, suggesting they were *bona fide* EV particles [22,30]. On the contrary, DiI-negative events were not significantly affected by incubation with NP-40, suggesting they were not membrane-enclosed particles (Figure 3B). The number of DiI-positive events detected in control samples such as buffer, unconditioned medium, or medium conditioned by unstained cells was two orders of magnitude lower than the number of DiI-positive particles in the cleared conditioned medium from stained cells (Figure 3C). Altogether, these data suggest that the conditioned medium of DiI-stained cells contains fluorescent objects that fulfill the definition of EVs and that can be efficiently detected by IFC. Importantly, the number of DiI positive particles detected by IFC in the cleared conditioned medium of stained cells correlated significantly well with the DiI fluorescence detected by spectrofluorometry (Figure 3D). Similar results were obtained with DiD-labeled EVs through a wide range of EV concentrations (Appendix A). Therefore, spectrofluorometry could be used as an approach for fast screening of changes in EV secretion by fluorescently labeled cells.

In the experiments described above, we used a 488 nm laser to detect labeled EVs, and a 785 nm laser for side scatter (SSC) detection. Although it was proposed that the use of a 405 nm laser might have advantages over the 785 nm laser [22], in our experiments we did not find that to be the case, as the 405 nm laser only increased the detection of background noise (Appendix A).

### 3.2. Cell Culture Conditions

Careful consideration should be given to the medium in which the cells are incubated during EV collection to prevent potential confounding effects arising from the presence of serum-derived EVs and serum nanoparticle components [31]. As have others before us [32], we found that even established protocols that involve ultracentrifugation at 100,000× *g* for 16 h to deplete EVs from serum (EV-depleted serum) result in a significant amount of EVs still remaining in the serum (Appendix A). Therefore, unless otherwise specified, in our studies, EVs secreted by N2a cells were collected in medium that did not contain serum. To prevent possible effects of serum depletion on cells that could potentially affect EV secretion [17,33,34] the culture medium used for the collection period was carefully selected to allow optimal cell survival and to minimize the effects of the switch to nutrient-poor medium on EV secretion, including those caused by changes in autophagy. To this end, we compared cells grown in regular culture medium (DMEM:OptiMEM, 1:1 with sodium pyruvate and L-glutamine supplements, as indicated in Materials and Methods) containing 10% EV-depleted fetal bovine serum (EVd), with cells grown in DMEM:OptiMEM (1:1) with sodium pyruvate and L-glutamine supplements alone (Opti) or supplemented with N-2 or B27 supplements. N2a cell viability was not significantly affected by the lack of serum in the medium (Appendix A). However, only cells grown in DMEM:OptiMEM + N-2 supplement (N-2 medium) displayed similar metabolic activity (Appendix A), autophagic activity (Appendix A), and secretion of EVs (Appendix A) compared to cells grown in serum-containing medium. Therefore, N-2 medium was used for EV collection in all experiments. All parameters described here, including the use of alternative culture media should be tested and validated for each different cell type.

### 3.3. Comparative Analysis of EVs in the Cleared Conditioned Medium and upon Separation by Ultracentrifugation or Ultrafiltration-Size Exclusion Chromatography

Next, we sought to determine whether the fraction used for EV analysis and the method of EV separation affect EV recovery or skew the analysis towards specific subpopulations, based on size and EV markers. In particular, we compared data obtained from the analysis by IFC of EVs in the cleared conditioned medium, with data obtained after EV separation with established and widely used methods, such as ultracentrifugation at 100,000× *g* (UC) and ultrafiltration-size exclusion chromatography (SEC), following the protocol illustrated in Figure 1. Transmission electron microscopy of EVs separated by UC (Pellet_100K_ ) or SEC (SEC_peak_) demonstrated, as expected, the presence of characteristic cup-shaped EVs in both preparations (Figure 4A). The Pellet_100K_ also contained several EV aggregates, a common technical artifact of UC procedures [33]. The immunoblots in Figure 4A show quality controls for the cleared conditioned medium_,_ the Pellet_100K_, and the SEC_peak_, including the presence of the EV marker CD9 and the absence of calnexin, which indicates lack of contamination from apoptotic bodies and cell debris. The procedure of SEC successfully separated EVs from soluble proteins, and DiI fluorescence was exclusively present in the EVs fraction, as expected for EVs labeled in-cell (Figure 4A). Both UC and SEC procedures for EV separation resulted in very low yields, as indicated by the significant decrease in the number of particles detected by IFC and nanoparticle tracking analysis (NTA) after UC and SEC, compared to the original unprocessed cleared conditioned medium (Figure 4B). Compared to the latter, both methods indicate that less than 10% of EVs were recovered in the Pellet_100K_ after UC, mainly because a significant proportion of EVs remained in the supernatant. Similarly, the number of EVs recovered by SEC (SEC peak) corresponded to only 12.6 ± 0.99% (by IFC) and 20.12 ± 6.23% (by NTA) of the total EVs in the cleared conditioned medium. However, the EV particle size distribution as well as mode size assessed by NTA were not significantly affected by the sample or isolation technique used (Figure 4C).

Given the remarkable loss of EVs resulting from UC and SEC separation procedures, we investigated whether the EVs recovered by UC are representative of the total EV population present in the unprocessed cleared conditioned medium, or of distinct EV subpopulations that might preferentially pellet at 100,000× *g*. IFC facilitates the detection of EV subpopulations in a heterogeneous sample [21]. To label distinct EV subpopulations, we incubated the cleared conditioned medium with anti-CD9 antibodies and Annexin V (which binds to phosphatidylserine, PS) prior to EV separation by UC. Based on these markers, IFC analysis of DiI-labeled particles showed the presence of four different EV populations in the cleared conditioned medium and in the Pellet_100K_ (Figure 5): DiI^+^/CD9^−^/Annexin^-^-EVs represented the larger population (around 60% of the total), followed by DiI^+^/CD9^+^/Annexin^−^-EVs (~20%), with the remaining approximately 20% divided between DiI^+^/CD9^−^/Annexin^+^-EVs and DiI^+^/CD9^+^/Annexin^+^-EVs. These four populations were equally represented in the cleared conditioned medium and in the Pellet_100K_ (Figure 5 and Appendix A).

### 3.4. Direct Analysis of EVs in the Cleared Conditioned Medium after Cell Treatment with Bafilomycin

To assess the suitability of the direct analysis by IFC of the cleared conditioned medium from DiI-labeled cells to study the regulation of EV secretion, we used bafilomycin (Baf), a compound that blocks autophagic flux and consequently increases EV secretion [34,35,36]. We selected the minimum concentration of Baf required to achieve maximum autophagic blockade in N2a cells (Appendix A). EVs were collected from DiI-labeled N2a cells treated with Baf (150 nM) or vehicle for 4 h and analyzed by IFC directly in the cleared conditioned medium, or in the Pellets_100K_ after separation by UC. Representative dot plots are shown in Appendix A. Analysis of the cleared conditioned medium, but not the Pellet_100K_, revealed the expected increase in the number of EVs secreted upon cell treatment with Baf (Figure 6A). Moreover, only the analysis performed in the cleared conditioned medium allowed to detect small but significant changes in the relative abundance of different EV subpopulations, with an increase in DiI^+^/CD9^−^/Ann^−^ EVs (Figure 6B and Appendix A).

For some functional studies such as those that involve cellular uptake of EV, the isolation of EVs from the CCM is unavoidable. Importantly, in-cell labeling of EVs allows the detection of EV internalization by cultured cells (Appendix A).

## 4. Discussion

### 4.1. EVs Labeling

EVs labeling with a pan-marker is a current challenge in the field. Our simple and versatile protocol of in-cell labeling with the lipophilic dyes DiI and DiD labeled both plasma and intracellular membranes of N2a cells. Although a decrease in cell labeling was observed at the end of the experiment, it was not reflected by a detectable decrease of EV labeling, indicating that prelabeling of cells with DiI results in the stable incorporation of the dye into EVs, and allows for reliable EV detection by IFC in a timeframe that is suitable for most cell-based studies. However, we cannot exclude the possibility that vesicles harboring low levels of DiI would be missed using this protocol for EV labeling and detection. We successfully used this indirect EV labeling method with DiI or DiD in different cell types in culture and using both DiI or DiD in the same cell type.

Using NTA we identified EVs of varying size, suggesting that DiI labels EV of different cellular origin. We found a strong positive correlation between the number of DiI- or DiD-labeled objects detected by IFC in the cleared conditioned medium and the DiI or DiD fluorescence assessed by spectrofluorometry, suggesting that measurements of fluorescence in the cleared conditioned medium from cells stained with these lipophilic dyes would provide a rapid estimation of EV secretion.

Many current strategies to label EVs released by cultured cells are based on labeling EVs in vitro after separation, or labeling parental cells, which results in the release of fluorescent EVs. Methods to label EVs in vitro have used many compounds including amine reactive dyes (carboxyfluorescein succinimidyl ester CFSE) [23,37], lipophilic dyes (PKH dyes, DiI, DiD, DiR, etc.) [19,23,24,37,38,39,40,41] and Mem dyes [42] among others [43]. Labeling of EVs in vitro suffers from the limitations inherent to the method employed for EV separation prior to labeling (discussed below) in addition to specific complications that arise during the labeling procedure, depending on the fluorophore employed. Dye aggregation has been reported during the labeling process using PKH dyes and DiI, which makes EV labeling in vitro with those lipid dyes unreliable, unless rigorous controls are used to discriminate between labeled EVs and non-EV-fluorescent particles (micelles, nanoparticles or aggregated dye) [23,24,25,26]. In addition, labeling EVs with PKH may cause a size shift towards larger size vesicles potentially through PKH nanoparticles fusion/aggregation [44]. Moreover, the procedure requires dye elimination after labeling and the methodology used for that purpose also plays a significant role in the final product [24]. Labeling in vitro with CFSE prevents artifacts due to dye aggregation and does not affect the size of EVs [44], but although some studies suggested that removal of the unbound CFSE is not required [37], others detected increased fluorescence in the background noise, possibly as a result of spontaneous hydrolysis of free dye [23]. Similarly, no large aggregation and no significant change of the apparent size of EVs were observed with the Mem dyes, although excess dye needs to be eliminated after labeling [42].

Labeling of parental cells with different compounds such as CFSE [45], PKH dyes [19,46], DiI, and DiD [39] have produced mixed results. CFSE may be toxic to the cells at the concentration required to label EVs; vesicles may not harbor enough CFSE fluorescence to be detected by flow cytometry [19] and only a small subfraction of the total EV population secreted, corresponding to the pellet after 10,000× *g* centrifugation, is labeled [45].

In our method, even though labeling was performed at saturating concentrations of DiI (based on manufacturer’s instruction) we did not observe cytotoxicity due to cell over-labeling as reported for other chromophores [47]. Because the cells are labeled in bulk, before seeding them in culture, all experimental groups are labeled equally. Moreover, our method offers a simple solution to determine if different cell types or different variants (e.g., wild-type and mutant cells) are labeled differently under similar conditions.

Another strategy to label EVs in cell involves expression of fluorescent reporters fused to EV-specific proteins markers [48,49,50] or carrying farnesylation or palmitoylation consensus sequences [25]. In the first scenario, there have been reports on the low proportion of EVs that are labeled [38] as well as the fact that labeling may be restricted to selective subpopulations of EVs, limiting the observations to only a few subtypes of EVs. The expression of membrane-bound fluorescent proteins overcomes the latter limitation, however the fluorescence intensity of EVs depends on protein expression level on the EV membrane, and it has been suggested that the expression of fluorescent proteins may affect EV properties and cargos. [51]. In summary, most current approaches for EVs labeling present limitations and there is a recognized need for better pan-EV dyes [26]. Recently, metabolic labeling of EVs with azido sugars- or phospholipid-based biorthogonal conjugation have been reported, mostly to study in vivo biodistribution of EVs [52,53]. Although these strategies would label all EV populations independently of their cellular origin and would not influence EV proteins nor affect the structural integrity of EVs, the labeling protocols still require the separation of the excess fluorescent dye following the in vitro click chemistry reaction.

While optimization of techniques for labeling EVs in vitro is important for the analysis of EVs in biological samples and/or to accurately follow EV biodistribution in vivo, we report here a simple method to label EVs released by cultured cells that can be reliably used for the study of EV secretion regulation.

### 4.2. EVs Separation and Analysis

EVs separated by UC and SEC were analyzed in parallel with EVs present in the cleared conditioned medium before separation, using IFC and NTA in a complementary approach. Both NTA and IFC provide the concentration of particles in a sample. NTA works by relating Brownian motion to particle size to determine number and size [54], while IFC works by detecting fluorescent energy from dyes or probes bound to EVs. IFC has addressed many of the limitations of traditional flow cytometry for measuring EVs with the added advantage of imagery confirmation [22,55,56], but does have its own limitations as well (necessity of fluorescent signal for small particle detection). Similar data would be expected from any high-resolution flow cytometry platform that can resolve extracellular vesicles.

Our analysis confirmed several issues associated with the separation of EVs by UC or SEC. In agreement with previous reports [57], NTA detected significantly higher numbers of particles than IFC in the same samples, because it would also detect non-EV particles such as large protein aggregates and dust [58], as well as EVs with very low DiI fluorescence that might escape detection by IFC. Despite these differences, both IFC and NTA indicated very poor EV yield using UC and SEC as shown before [55,56]. Although NTA analysis agreed with earlier indications that UC causes more significant reduction of particle yield than SEC [27], analysis by IFC showed no significant differences in total number of EVs isolated by UC and SEC and there was no difference in the levels of CD9 detected in EVs separated by either method. An additional disadvantage of EV separation by UC is the artificial aggregation and/or fragmentation of EVs, which might lead to artifacts during single particle analysis and flow cytometry analysis [59] and may mask antigens on the EV surface, thus complicating phenotypic analysis of EVs based on markers [33]. Furthermore, aggregation and co-sedimentation of free proteins with the EV pellet may cause EV contamination [60] that may remain undetected depending of the method employed for EVs analysis. Despite the significant loss of EVs upon isolation by UC, the proportion of DiI-detectable vesicles in each of the four EV populations analyzed in our studies was unchanged, although we cannot exclude that other EV subpopulations might be affected by UC.

Our studies demonstrated for the first time that direct analysis of the cleared conditioned medium may allow to detect changes in EV secretion that might be otherwise missed if EVs are isolated by UC, at least when the changes are of a relatively small magnitude. In our experiments with bafilomycin, the expected increase of EV release was detected by IFC analysis of the cleared conditioned medium, while analysis of the pellet after UC provided non-significant differences in EV secretion. Moreover, the relative abundance of EV subpopulations in the cleared conditioned medium revealed subtle drug-induced changes that could not be detected in the Pellet_100K_. Therefore, all considered, we propose that for the study of EV secretion, direct analysis of EVs in the cleared conditioned medium is preferable to the analysis in the pellet 100K, unless, of course, maximum EV concentration or separation from soluble components in the medium (proteins and other factors) are required for downstream applications. Moreover, analysis of the cleared conditioned media should be an integrated component of any EV study so as to reduce the chance of losing valuable data due to technical artifacts.

## 5. Conclusions

In summary, we developed a protocol that makes use of a unique combination of in-cell-labeling and EV analysis by IFC directly in the cleared conditioned medium. Altogether, the data presented here indicate that cell prelabeling with DiI, coupled with IFC analysis of fluorescent EV particles directly in the cleared conditioned medium represents a powerful and convenient method to study EV secretion in vitro, with minimal sample handling and loss.

Although IFC analysis has the added advantage of imagery confirmation [22,55,56] compared to classic flow cytometry, any high-resolution flow cytometry platform that can resolve EVs could be used instead of IFC to analyze EVs in the conditioned medium.

Our protocol prevents selective loss of EV subpopulations that can occur with SEC- and UC-based protocols and, therefore, allows a more accurate characterization of EV subpopulations secreted by cells in the conditioned medium. Therefore, we propose that analysis of the cleared conditioned media should be an integrated component of any EV secretion study so as to reduce the chance of losing valuable data due to technical artifacts.

## Figures and Tables

**Figure 1 cells-11-00351-f001:**
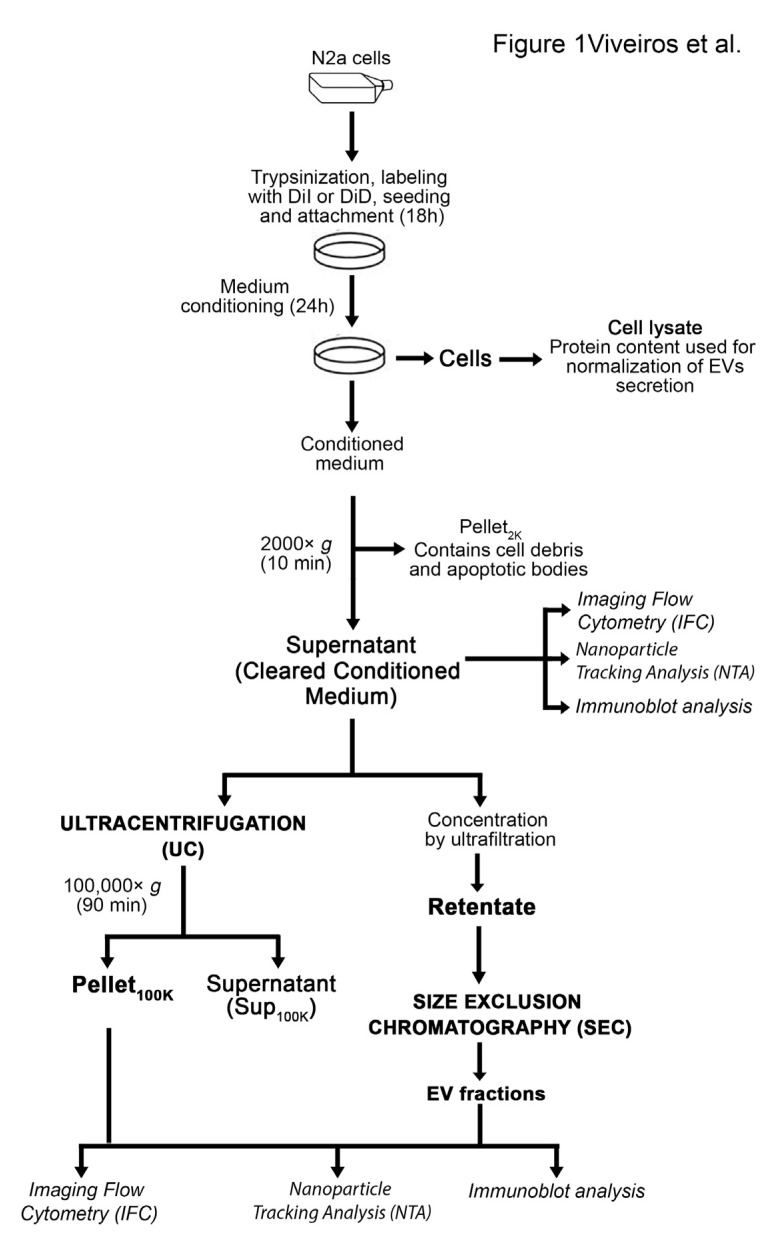
Outline of procedures for EV labeling, collection, and separation. Techniques used to analyze EV samples are indicated in italic.

**Figure 2 cells-11-00351-f002:**
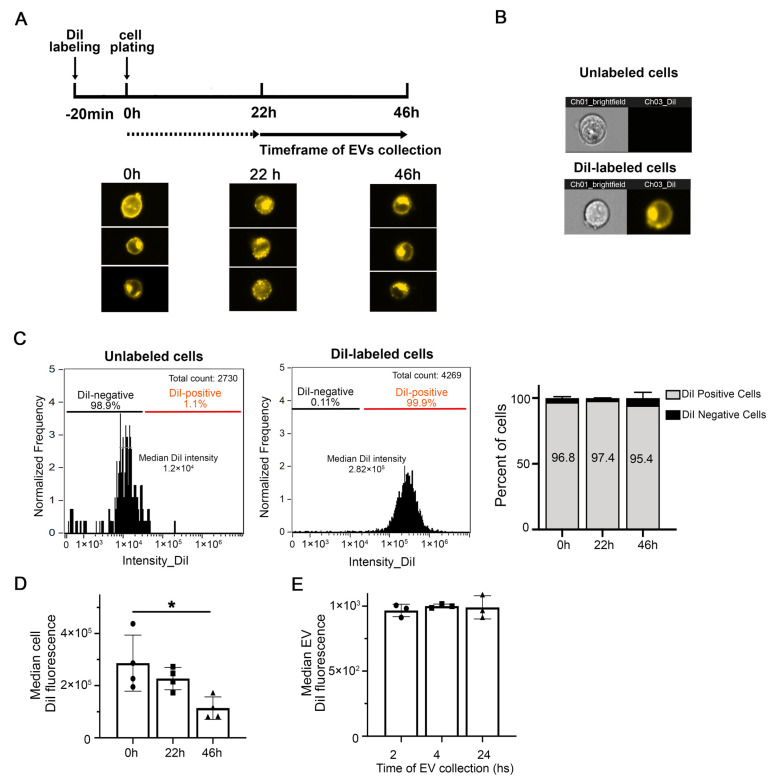
DiI effectively labels cells in culture. (**A**) Time-course of fluorescent cell analysis and EV collection after N2a cell labeling with DiI. Representative IFC images of cells immediately after labeling with DiI (0 h) and 22 h and 46 h postlabeling. DiI fluorescence is distributed throughout the cells, at the membrane and in intracellular compartments. EVs were collected in the conditioned medium between 22 and 46 h after DiI staining. (**B**) Representative IFC images of unlabeled (top) and DiI-labeled N2a cells (bottom). Images taken in the brightfield and DiI channels are shown. (**C**) Representative histogram of DiI cell intensity for unlabeled (left) and labeled cells (right). The bar graph on the right shows that the labeling procedure results in over 95% of cells that are DiI-positive up to 46 h following cell staining. (**D**) The median DiI intensity of the cells is not significantly affected after 22 h in culture but is decreased at 46 h after cell staining. Columns are means +/− SD of four experiments. Small triangles, circles and squares indicate values from individual experiments. One-way ANOVA followed by Dunnett’s multiple comparisons test was performed using GraphPad Prism * *p* < 0.05. (**E**) The median DiI fluorescence of EVs detected in the cleared conditioned medium does not significantly change at 24, 28, and 46 h after cell staining (corresponding to 2, 4, and 24 h collection). Columns are means +/− SD of four experiments. Small triangles, circles and squares indicate values from individual experiments One-way ANOVA followed by Dunnett’s multiple comparisons test was performed * *p* > 0.5.

**Figure 3 cells-11-00351-f003:**
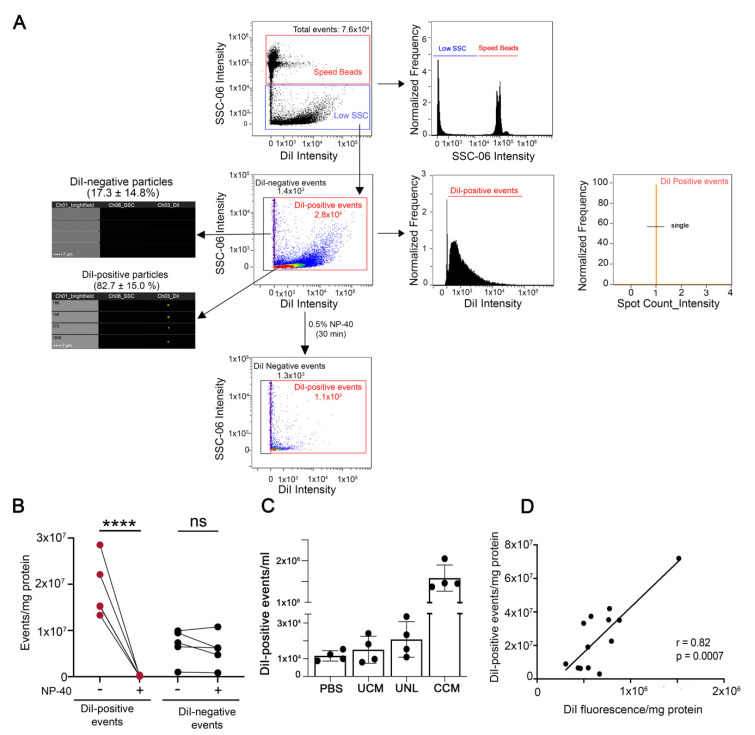
IFC allows the detection of EVs released in the cleared conditioned medium by DiI-labeled cells. (**A**) Gating strategy for EV analysis. First samples were gated on intensity in the scatter channel (Ch06 (SSC) to remove any remaining SpeedBeads from the analysis. A representative density dot plot of the low scatter population is included, together with images taken in the brightfield, side scatter (SSC), and DiI channels of DiI-negative and DiI positive events. The majority of events are DiI-positive and disappear (are lysed) upon sample incubation with the detergent NP-40 (lower dot plot), demonstrating that they are membrane-enclosed particles. The histogram of DiI intensity on the right determines the population of DiI-positive events. To ensure the analysis of only 1 EV in each image, a mask was created to identify DiI intensity (Intensity (M03, Ch03_DiI, 50-4095)). Using this mask, we developed a spot-count feature (Spot Count_Intensity (M03,Ch03_DiI, 50-4095)_4) to gate on images that had no more than 1 DiI spot (single). (**B**) Quantification of the effects of NP-40 on DiI-positive and DiI-negative particles in four independent experiments. While DiI-positive particles are highly sensitive to lysis by NP-40, DiI-negative particles are not, indicating they are not EVs. One-way ANOVA followed by Sidak’s multiple comparisons test was performed. **** *p* < 0.001, ns = not significant. (**C**) Quantification of DiI-positive events detected in buffer (PBS), unconditioned medium (UCM), medium conditioned by unlabeled cells (UNL) and cleared conditioned medium (CCM) from DiI-stained cells in four independent experiments. All samples were subjected to the same procedures, including the centrifugation at 2000× *g* required to obtain the CCM, and were measured for the same time. (**D**) Pearson’s correlation analysis between the number of DiI-positive events detected by IFC and DiI fluorescence measured by spectrofluorometry in the cleared conditioned medium.

**Figure 4 cells-11-00351-f004:**
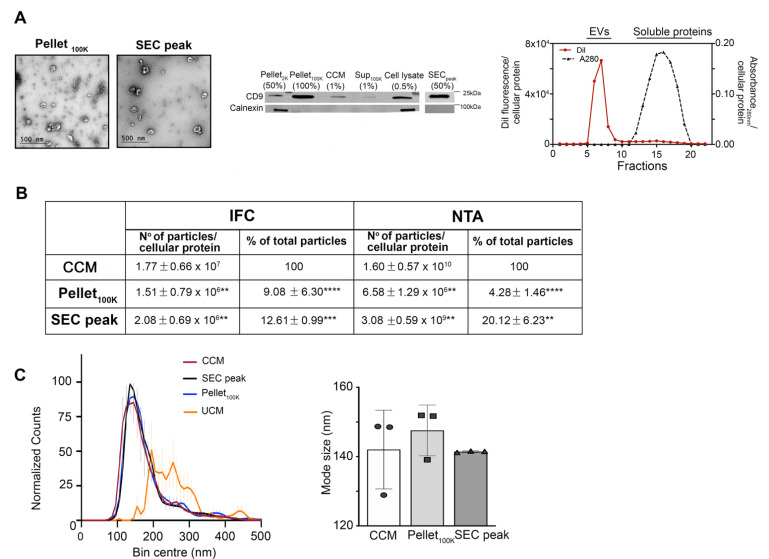
Characterization of EVs separated by sequential ultracentrifugation or ultrafiltration size-exclusion chromatography. EVs were separated by UC and SEC as indicated in Figure 1. (**A**) Electron micrographs of Pellet_100K_ and SEC_peak_ fractions show characteristic cup-shaped EV particles. The immunoblot shows the presence of the EV marker CD9 and the absence of calnexin in all fractions obtained from the conditioned medium after the initial centrifugation at 2000× *g*, indicating the fractions are not contaminated by apoptotic bodies or cell debris. Numbers in parenthesis indicate the fraction for each sample that was loaded in the gel. A representative chromatogram of EV separation by SEC is shown on the right. (**B**) Summary of the average number of particles present in each fraction (normalized for cellular protein content) and percentage relative to the total number of particles detected in the cleared conditioned medium (CCM). Particle numbers were determined by imaging flow cytometry (IFC) and nanoparticle tracking analysis (NTA). Values are means ± SD of 6 (IFC) or 3 (NTA) independent experiments. The paired *t*-test was applied to compare the Pellet_100K_ or the SEC peak to the cleared conditioned medium using GraphPad Prism **** *p* < 0.0001, *** *p* < 0.001, ** *p* < 0.01 (**C**) Particle size distribution profiles of EVs in the cleared conditioned medium (CCM, circles) and after isolation by UC (Pellet_100K_, squares) or SEC (SEC peak, triangles), as detected by NTA. Each profile was obtained using the mean values from three experiments. Vertical lines are SD for each size bin. The graph on the right shows the mode size for each indicated fraction. Particle size is not significantly different among fractions. Data are mean values ± SD of three independent experiments. CCM = cleared conditioned medium, UCM = unconditioned medium.

**Figure 5 cells-11-00351-f005:**
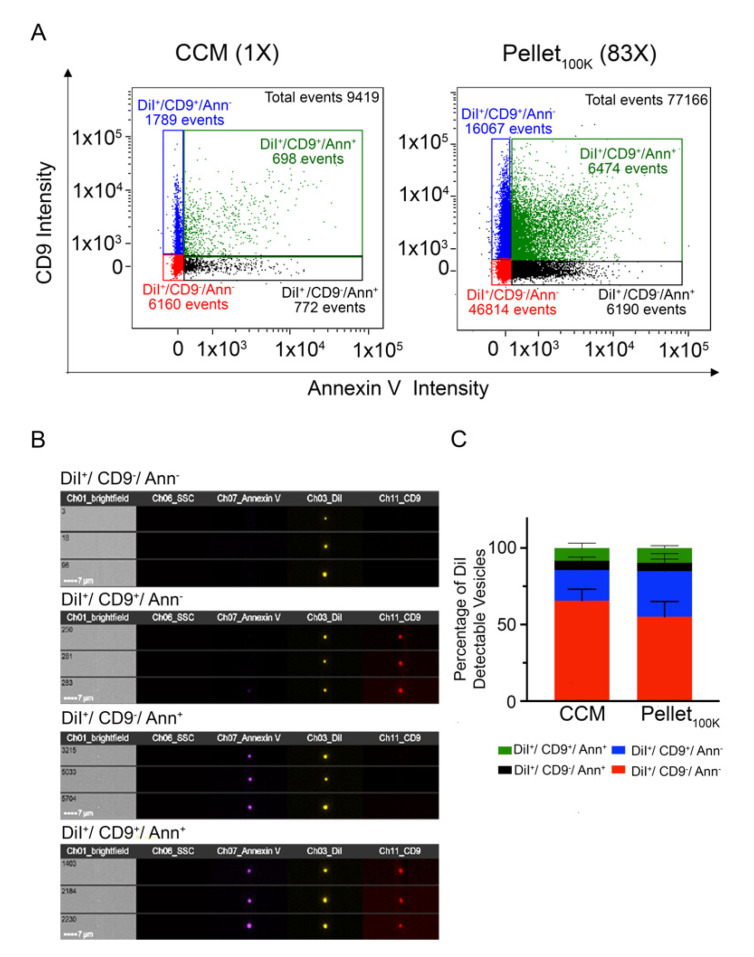
Comparison of markers distribution between the cleared conditioned medium and the Pellet_100K_. (**A**) Representative IFC dot plots for EVs labeled in vivo with DiI and in vitro with anti-CD9 antibodies and/or Annexin V. Analysis were performed in cultured conditioned medium (CCM) and the pellets that resulted from UC separation (Pellet_100K_). Numbers in parenthesis indicate the concentration factor in each sample. All samples were analyzed for the same time (10 min) in the same conditions. (**B**) Representative IFC images of particles in each subpopulation. (**C**) Quantification of the relative abundance of differentially labeled EV subpopulations in the cleared conditioned medium and in the Pellet_100K_. Data are mean percent values of detectable vesicles ± SD of four independent experiments. Paired *t*-test and estimation plot analysis were performed using GraphPad Prism No significant differences between groups were found.

**Figure 6 cells-11-00351-f006:**
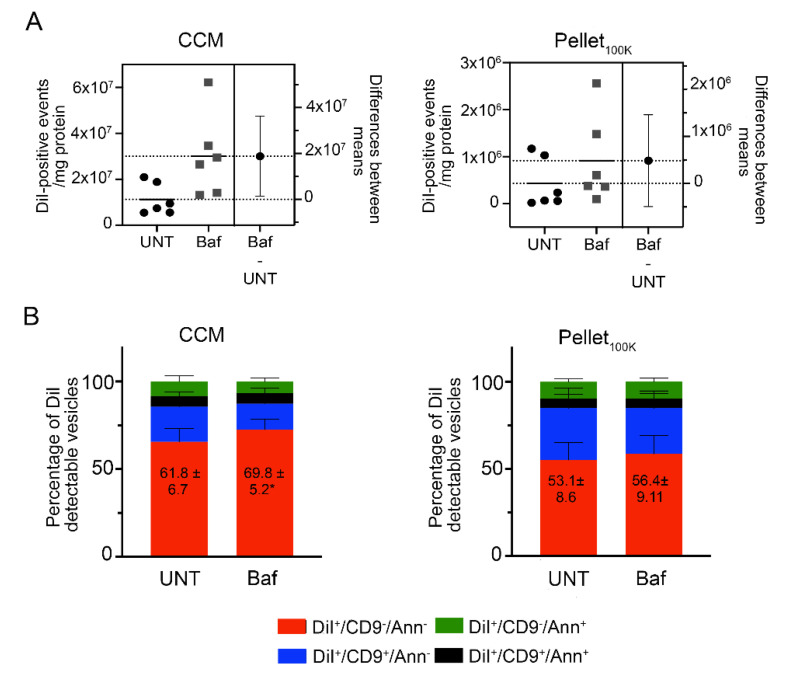
Effect of bafilomycin on EV release measured in the cleared conditioned medium and in the Pellet_100K_. EVs were collected from DiI-labeled N2a cells treated with Baf (150 nM) or vehicle (UNT) for 4 h and analyzed in the cleared conditioned medium (CCM) or after isolation by UC. The cleared conditioned media were adjusted based on the protein content of donor cells before isolation by UC and analysis by IFC. All samples were counted for the same time (10 min). (**A**) The total number of DiI-positive EVs in the cleared conditioned medium and in the Pellet_100K_, respectively, was analyzed. Significance was determined by unpaired *t*-test and estimation plot analysis performed. The left panel in each plot shows data for individual experiments (circles: untreated; squares: treated with Baf) and their mean. The right panel shows the effect size (difference between means) and its 95% confidence interval. (**B**) Cleared conditioned medium and Pellet_100K_ preparations were labeled in vitro using anti-CD9 antibodies and Annexin V. Graphs show the relative abundance of differentially labeled EV subpopulations in each fraction. Data are mean percent values ±SD for three independent experiments. Significance determined by unpaired *t*-test and estimation plot analysis were performed. * *p* < 0.05.

## Data Availability

Not applicable.

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
