# Peer review of "In-Cell Labeling Coupled to Direct Analysis of Extracellular Vesicles in the Conditioned Medium to Study Extracellular Vesicles Secretion with Minimum Sample Processing and Particle Loss"

_cells, 2022, doi:10.3390/cells11030351_

Round 1

Reviewer 1 Report

In this manuscript, the authors reported a method for in-cell EV labeling. This novel protocol can reduce the EV sample processing and loss. Besides, it provides a simplified and reliable method to study EV secretion and regulation in cell models. In general, the experiments were well-designed and performed. The conclusion is well supported by the results as well. The manuscript is well written and whilst it shows an interesting method for researchers. One question for this method is that cell-free EV labeling has been widely used for tracking EV internalization in vitro or EV distribution in vivo. It would be important and helpful to compare cell-free and in-cell EV labeling in these applications. For example, if in-cell labeled EVs are detectable after internalization in vitro or administrated in vivo.

Author Response

In this manuscript, the authors reported a method for in-cell EV labeling. This novel protocol can reduce the EV sample processing and loss. Besides, it provides a simplified and reliable method to study EV secretion and regulation in cell models. In general, the experiments were well-designed and performed. The conclusion is well supported by the results as well. The manuscript is well written and whilst it shows an interesting method for researchers. One question for this method is that cell-free EV labeling has been widely used for tracking EV internalization in vitro or EV distribution in vivo. It would be important and helpful to compare cell-free and in-cell EV labeling in these applications. For example, if in-cell labeled EVs are detectable after internalization in vitro or administrated in vivo.

Response:

We thank the reviewer for their comments. As part of our studies we had tested the internalization of EVs labeled in-cell and found that they are detectable after internalization by cells in culture. We have added a statement on lines 474-476 and show the correspondent data in Supplementary Figure 14.

Reviewer 2 Report

This manuscript describes a simple, well conducted study to evaluate a method for in-cell EV labeling with fluorescent lyophilic dyes coupled with direct characterization of secreted EVs in the conditioned medium. Even if the procedure per se is not original, it provides interesting observations on the phenotypes of freshly secreted EVs, compared to EVs isolated by ultracentrifugation or by size-exclusion chromatography.

The Authors should however define more critically the possible applications of their procedure. Stating that the described method “… should be an integral component of any EV study...” is too much. For instance, a study on EVs isolated and concentrated for therapeutic use should concentrate on the characteristics of the final product, rather than of the original medium. This is especially true since isolation methods invariably result in the loss of most EVs, as described in the paper.

Minor observations

Lines 177-178: “… blocked with Tween 80 as previously described.” Please add a reference or a more detailed description

Line 472: should refer to Supplementary Fig. 12 (not 13)

Line 537: “in vivo” labeling should be corrected with “in cell” labeling.

Author Response

Point 1: The Authors should however define more critically the possible applications of their procedure. Stating that the described method “… should be an integral component of any EV study...” is too much. For instance, a study on EVs isolated and concentrated for therapeutic use should concentrate on the characteristics of the final product, rather than of the original medium. This is especially true since isolation methods invariably result in the loss of most EVs, as described in the paper.

Response: Throughout the paper we indicate that the main application of our procedure of labeling and analysis of the conditioned medium is for the study of EV secretion. The concept is included in the title of the paper as well as in the conclusion (line 617 of revised manuscript). We agree that our EV labeling protocol and analysis might not apply to EVs from origins different than cultured cells or for EVs derived from cultured cells but to be used for other functional studies. The statement has been modified to clarify that it applies to EV secretion. In the current version it reads “….analysis of the cleared conditioned media should be an integrated component of any EV secretion study so as to reduce the chance of losing valuable data due to technical artifacts.” (line 626).

Point 2:  177-178: “… blocked with Tween 80 as previously described.” Please add a reference or a more detailed description

Response: a reference has been added (line 178- reference 28- Lee, K. J. et al. Modulation of nonspecific binding in ultrafiltration protein binding studies. Pharm Res 20, 1015-1021, doi:10.1023/a:1024406221962 (2003))

Point 3: Line 472: should refer to Supplementary Fig. 12 (not 13)

Response: the Supplementary figure referred on line 472 (current line 473) is on the different EV populations measured in the cleared conditioned medium and in the Pellet100 and corresponds to Supplementary Fig. 13. Supplementary Fig. 12 represents the dotplots of EVs collected from cells treated with and without Baf and is indicated in line 468.

Point 4: Line 537: “in vivo” labeling should be corrected with “in cell” labeling.

Response: The term has been changed to “in cell” – line 546 of current version